# Bacterial Community with Plant Growth-Promoting Potential Associated to Pioneer Plants from an Active Mexican Volcanic Complex

**DOI:** 10.3390/microorganisms10081568

**Published:** 2022-08-04

**Authors:** Clara Ivette Rincón-Molina, Esperanza Martínez-Romero, José Luis Aguirre-Noyola, Luis Alberto Manzano-Gómez, Adalberto Zenteno-Rojas, Marco Antonio Rogel, Francisco Alexander Rincón-Molina, Víctor Manuel Ruíz-Valdiviezo, Reiner Rincón-Rosales

**Affiliations:** 1Laboratorio de Ecología Genómica, Tecnológico Nacional de México, Instituto Tecnológico de Tuxtla Gutiérrez, C.P., Tuxtla Gutierrez 29050, Chiapas, Mexico; 2Centro de Ciencias Genómicas, Universidad Nacional Autónoma de México, Av. Universidad s/n, Col. Chamilpa, C.P., Cuernavaca 62210, Morelos, Mexico; 3Departamento de Investigación y Desarrollo, 3R Biotec SA de CV, C.P., Tuxtla Gutierrez 29000, Chiapas, Mexico

**Keywords:** active volcano, bacterial communities, pioneer plants, PGPB, inoculation

## Abstract

Microorganisms in extreme volcanic environments play an important role in the development of plants on newly exposed substrates. In this work, we studied the structure and diversity of a bacterial community associated to *Andropogon glomeratus* and *Cheilanthes aemula* at El Chichón volcano. The genetic diversity of the strains was revealed by genomic fingerprints and by 16S rDNA gene sequencing. Furthermore, a metagenomic analysis of the rhizosphere samples was carried out for pioneer plants growing inside and outside the volcano. Multifunctional biochemical tests and plant inoculation assays were evaluated to determine their potential as plant growth-promoting bacteria (PGPB). Through metagenomic analysis, a total of 33 bacterial phyla were identified from *A. glomeratus* and *C. aemula* rhizosphere samples collected inside the volcano, and outside the volcano 23 bacterial phyla were identified. For both rhizosphere samples, proteobacteria was the most abundant phylum. With a cultivable approach, 174 bacterial strains were isolated from the rhizosphere and tissue of plants growing outside the volcanic complex. Isolates were classified within the genera *Acinetobacter, Arthrobacter, Bacillus, Burkholderia, Cupriavidus, Enterobacter, Klebsiella, Lysinibacillus, Pantoea, Pseudomonas, Serratia, Stenotrophomonas* and *Pandoraea*. The evaluated strains were able to produce indole compounds, solubilize phosphate, synthesize siderophores, showed ACC deaminase and nitrogenase activity, and they had a positive effect on the growth and development of *Capsicum chinense*. The wide diversity of bacteria associated to pioneer plants at El Chichón volcano with PGPB qualities represent an alternative for the recovery of eroded environments, and they can be used efficiently as biofertilizers for agricultural crops growing under adverse conditions.

## 1. Introduction

After a volcanic eruption disturbance, the terrestrial ecosystem undergoes primary succession, where the newly formed or exposed land surfaces comprise new parent materials (e.g., tephra, ash, lava), rather than developed soils [1]. In this scenario, the soil substrates are formed mainly of sulfide-rich minerals generating sulfuric acid through oxidation processes, thus the soil shows low pH [2]. As a result, a lack of soil organic matter and the diminution of critical bioavailable nutrients shape and limit the ecosystem development in volcanic zones [3]. True primary succession after a volcanic disturbance is rare, and secondary succession is frequent. The main characteristics in this natural process are the presence of surviving organisms and a more or less mature soil containing an established seed bank and vegetative propagules [4]. Then pioneer plants, which establish symbiotic interactions with bacterial communities, colonize the newly formed soil and both collaborate in soil regeneration [5,6]. To achieve sustainability, ecosystem stability is an essential part where the early interactions between rhizosphere bacteria and pioneer plants play crucial roles in being able to avoid or tolerate extreme environmental conditions and nutrient limitations, and modify the physical and chemical characteristics of the substrates through various strategies [7,8]. Bacteria associated to plants promote plant growth through direct or indirect biochemical mechanisms, such as N_2_ fixation, inorganic phosphate solubilization, auxin synthesis, and production of siderophores and other metabolites. The bacteria that show these biological qualities are known as plant growth-promoting bacteria (PGPB) [9]. Thereby, extremophile bacteria with plant growth-promoting properties in association with plants may help plant growth and adaptation under harsh environments. Recently, some studies related to rhizospheric and endophytic bacterial communities associated with pioneer plants in active volcanic environments have been published [10,11]. Nowadays, genetic mapping with molecular techniques such as next generation sequencing (NGS) technologies provides a unique view of the richness, composition and structure of the microbial community, essential to have an idea of the rhizospheric soil biological status in extreme conditions [12].

El Chichón is an active volcano located in the northwest of Chiapas, Mexico [13]. The most recent eruption of El Chichón was recorded in 1982; a total area of 10 km around the volcano was destroyed, causing significant damage to the native vegetation [14]. After the eruptive process, a 1.4 × 10^5^ m^2^ crater-lake was formed; actually, it is classified as a hydrothermal system [15]. This volcanic area has shown signs of recovery, mainly in the structure and conformation of the soil through ecological succession processes and is once again being colonized with some pioneer plant species. El Chichón volcano has been the subject of several investigations focused mainly on geophysical characteristics [14,16,17]. To date, there are few reports related to the diversity and structure of the bacterial community associated to pioneer plants in this extreme environment. In preliminary work, we studied the structure and diversity of the microbial community in the volcanic sediments of El Chichón [18] and recently we published information related to the diversity of PGPB associated with two pioneer plants [19]. These reports generated scientific interest and have given rise to new specialized investigations where next generation sequencing tools are being employed to exploit microbial communities [20].

Based on the above-mentioned research and in order to continue the research at this volcanic environment, this work aims at studying the structure and diversity of rhizospheric and endophytic bacteria from pioneer plants that grow at this extreme volcanic complex and determining their potential as PGPB.

## 2. Materials and Methods

### 2.1. Biological Sampling

Rhizospheric soil and plant tissues were obtained from the pioneer plants *A. glomeratus* (Poaceae) and *C. aemula* (Pteridaceae). These plants grow inside (crater-lake) and outside (domme) of El Chichón volcano (17.36° N, 93.23° W; 1100 m.a.s.l.). Inside the crater, the samples were randomly collected in a minimum area of 400 m^2^ [19]. The outer samples were collected at two different points on a transect (T_1_ and T_2_) separated by a distance of 1.5 km (Figure 1). At each point, three complete specimens containing rhizospheric soil were collected per plant species. The plants were extracted following the method described by Schafer et al. [21].

### 2.2. Rhizospheric Soil Characterization

Soil samples were carefully collected from the rhizospheres of the two pioneer plant species growing outside the crater and finely sieved for subsequent analyses. 

#### 2.2.1. Physicochemical Analysis

Rhizospheric soil samples were physicochemically analyzed. The pH, electrical conductivity (EC), cation exchange capacity (CEC) and soil organic matter (SOM) were determined according to Rincón-Molina et al. [19]. In addition, the organic carbon and total nitrogen were determined using a FLASH 2000™ auto-analyzer. Total phosphorus was determined with the solubilization method of HNO_3_/HClO_4_. The same determinations were performed in a soil sample used as control, which was collected inside the volcanic complex. The control soil consisted of a soil without plants.

#### 2.2.2. Chemical Analysis of Rhizospheric Soil Minerals and Metals

The minerals and metals analyzed in this study only included samples obtained from the pioneer plants growing inside the volcano. For minerals detection, rhizospheric soil of each sample was ground in Agate mortar and placed in a polymethylmethacrylate sample holder for X-ray diffraction (XRD) analysis. The mineral phases were determined with a Bruker DV8 advance diffractometer, using a CuKα target X-ray tube and a radiation of 40–30 mA, with a scanning speed of 25 min at an angle of 5° to 80° in 2θ [22]. 

For the elements detection, the samples were dried under vacuum at 50 °C at a pressure of 0.05 MPa for 48 h in an oven. Then, the soil was crushed to a particle diameter of 0.050 mm and after size homogenization, acid digestion was carried out following the method 3050B recommended by the Environmental Protection Agency (EPA). Later, 200 mg of each sample was treated with 10.0 mL of HNO_3_, 1.0 mL of H_2_O_2_ and 1.0 mL of HCl, and then heated to reflux at 180 °C for 2 h until the total oxidation of the organic matter. Total metal concentrations were determined by inductively coupled plasma-optical emission spectrometry (ICP-OES) on an Optima 7000 PerkinElmer spectrometer [23].

### 2.3. Characterization of Bacterial Community Associated to Pioneer Plants

The structure and diversity of the bacterial community associated with the pioneer plants inside and outside El Chichón volcano were studied through the following techniques.

#### 2.3.1. Culture-Independent Characterization of Rhizospheric Bacteria

Total DNA was extracted from plant rhizospheric soils. This analysis was performed inside and outside the volcano using a commercial kit (ZymoBIOMICS™ DNA Miniprep). The DNA samples were sent to Macrogen Inc. (Seoul, Korea) for the amplification of the V3–V4 bacterial 16S rRNA variable region using the primers Bakt_341F: CCTACGGGNGGCWGCAG and Bakt_805R: GACTACHVGGGTATCTAATCC. Sequencing was performed using Illumina Miseq 2 × 300 paired-end [24]. The QIIME version 2.0 software pipeline was used to analyze the sequencing data [25]. Poor quality readings were eliminated from the data sets, i.e., a quality score < 25, containing homopolymers > 6, length < 400 nt, and containing errors in primers and barcodes. Operational taxonomic units (OTUs) were determined at a 97% similarity level with the UCLUST algorithm [26]. Chimeras were detected and removed from the data sets using the Chimera Slayer [27]. Sequence alignments were performed against the Greengenes core set (available from http://greengenes.lbl.gov/, accessed on 15 July 2020) and using representative sequences of each OTU using PyNAST, and filtered at a threshold of 75% [28]. Taxonomic assignation was performed with rarified data sets at 850 reads per sample to compare the same amount of sequences and using the naïve Bayesian rRNA classifier from the Ribosomal Data Project (http://rdp.cme.msu.edu/classifier/classifier.jsp, accessed on 20 July 2020) at a confidence threshold of 80% [29]. Heatmaps were constructed with the pheatmap package in R [30]. Raw sequence data were deposited in the Sequence Read Archive (SRA) database at the NCBI under the accession numbers SUB7748710 and SUB11308529.

#### 2.3.2. Culture-Dependent Characterization of Rhizospheric and Endophytic Bacteria

In a previous work [19], we reported the taxonomic characteristics of isolates obtained from plants growing inside the volcano. In this section, we complemented the study by characterizing isolates from the rhizosphere soil and plant tissues belonging to pioneer plants growing outside the volcano according to the procedure indicated by Rincón-Molina et al. [19]. DNA was extracted from bacterial isolates using kit for cells and tissues (Roche™). BOX patterns were identified by electrophoresis in 3.0% agarose gels and then richness (*d*) and diversity (*H*) index were calculated [31]. Isolates were identified by 16S rDNA gene amplification using universal primers 27f and 1492r [32]. The PCR products were digested with a restriction endonuclease Alu I (Promega™) to obtain genomic fingerprints using ARDRA (amplified ribosomal rDNA restriction analysis). PCR products were purified and Sanger sequenced (Macrogen™). Sequences were edited and assembled using BIOEDIT v7.2. Isolates were identified using the BLAST algorithm [33]. The new bacterial sequences were deposited in the GenBank database under the accession numbers OL843131 to OL843162 for isolates from outside the volcano.

### 2.4. Measurement of PGPB Efficiency

Different bacterial strains were grown in PY medium for 24 h at 28 °C and used in the different plant growth-promoting tests listed below.

#### 2.4.1. Inorganic Phosphate Solubilization 

The isolates were inoculated in NBRIP medium containing insoluble tricalcium phosphate (Ca_3_(PO_4_)_2_); the pH was previously adjusted to 7.0 [34]. Phosphate solubilizing bacteria colonies were recognized by clear halos after five days of incubation at 30 °C and the phosphate solubilization index (PSI) was calculated as described by Liu et al. [35].

#### 2.4.2. Indole Acetic Acid (IAA) Production

A single bacterial colony was streaked onto LB agar amended with 5 mmol L^−1^ L-tryptophan [36]. Plates were overlaid with sterile Whatman no. 1 filter paper and bacterial strains were grown for 72 h at 28 °C. After incubation, the paper was removed and treated with Salkowski’s reagent at 28 °C for 60 min. The IAA was identified by the formation of a red halo on the paper surrounding the colony.

#### 2.4.3. Acetylene Reduction Assay (ARA)

Strains were grown in N-free minimal semisolid medium. The culture and detection conditions for acetylene reduction activity measures were performed according to the methodology described by Navarro-Noya et al. [37]. 

#### 2.4.4. ACC Deaminase

An inoculum of 10^9^ cel mL^−1^ (OD_600_nm = 0.2) of each of the bacterial strains were inoculated in culture medium containing: 0.25 g K_2_HPO_4_; 0.05 g MgSO_4_.7H_2_O; 0.025 g FeSO_4_.7H_2_O; 0.25 g CaCO_3_; 0.05 g NaCl: 0.0012 g NaMoO_4_.2H_2_O; 2.5 g glucose; 3.75 g agar; 240 mL distilled water; and 0.03% of ACC as the sole source of nitrogen. The Petri dishes were incubated at 30 °C for 4 days. The development of bacterial colonies indicates the production of ACC deaminase by the isolates [38]. 

#### 2.4.5. Siderophore Production

The bacterial isolates were grown in CAS-agar medium [chromeazurol-S (CAS), iron (III) and hexadecyl trimethyl ammonium bromide (HDTMA)] at 28–30 °C for 5 days. The production of bacterial siderophore was detected by a color change from blue to a fluorescent orange surrounding the colonies [39].

#### 2.4.6. Exopolysaccharide (EPS) Production 

EPS was determined as described by Paulo et al. [40]. The evaluated strains were inoculated into sterilized filter paper discs of 5 mm Ø placed on the surface of PY modified culture medium (containing 2% yeast extract, 1.5% K_2_HPO_4_, 0.02% MgSO_4_, 0.0015% MnSO_4_, 0.0015% FeSO_4_, 0.003% CaCl_2_, 0.0015% NaCl, 1.5% agar and 10% sucrose, with pH adjusted to 7.5) followed by incubation at 30 °C for 48 h. A mucoid layer formed around the paper discs suggested EPS production. In order to confirm the presence of EPS, the mucoid layer was transferred to a tube containing 2.0 mL absolute ethanol. The EPS presence was confirmed by the formation of a precipitate.

### 2.5. Plant Inoculation Assays

Plantlets of pepper (*Capsicum chinense*) were transplanted to a polystyrene pot containing sterilized peat moss as substrate. Plants were inoculated with 2 mL of each selected strain at a concentration of 1 × 10^6^ CFU mL^−1^. Plants treated with KNO_3_-N served as positive control (fertilizer). Uninoculated plants were used as a negative control. Four replicates were used per treatment. The plants were grown under greenhouse conditions for 90 days following a completely randomized design. At harvest, total height, plant weight, root weigh, root length, stem diameter, number of fruits, chlorophyll content, total phosphorus, total nitrogen and organic carbon were determined. 

### 2.6. Statistical Analysis

ANOVA analysis was performed with an alpha level = 0.05. The Tukey test (*p* < 0.05) was performed on those variables that were significant (Statgraphics Centurion v.2015.1). The correlation between relative abundance of the bacterial groups at phyla and genera levels of each of the plant species and the physicochemical characteristics were explored with a principal component analysis (PCA) [41]. The PCA charts were obtained using the R Studio v4.1.1 platform, and for multivariate data analysis, the Factoextra for multivariate data analysis was employed.

## 3. Results

### 3.1. Rhizospheric Soil Physicochemical Characteristics 

The soil samples obtained from the rhizospheres of *A. glomeratus* and *C. aemula* had significant variations (*p* < 0.05) in relation to the different physicochemical parameters evaluated (Table 1). The rhizospheric soil pH from plants inside the volcano was moderately acidic (range from 5.1 to 6.7) compared to the soil pH from transect T_1_ (outside the volcanic complex), where the pH was strongly acidic (range from 5.4 to 4.8). The recorded pH value of the control soil was the most acidic (4.3). The electrical conductivity (EC) determined among the rhizosphere samples showed significant differences (*p* < 0.05). Inside the crater, the EC in *A. glomeratus* was higher (0.97 dSm^−1^) than that registered in *C. aemula* (0.48 dSm^−1^). Outside the volcanic complex, in transect T_1_ EC values were similar in both plants (0.02 dSm^−1^). In transect T_2_, the EC value was higher in *A. glomeratus* than in *C. aemula*. In control soil, a high EC value (0.91 dSm^−1^) was also determined. The highest CEC value was determined in the *C. aemula* rhizosphere located at transect T_2_ and a lowest value was registered in *A. glomeratus* plants growing in transect T_1_. A low CEC value was registered in samples of control soil. On the other hand, the amount of total N, organic C content and C:N ratio were significantly higher in the *C. aemula* rhizosphere (transect T_2_) compared to the other samples. The total P value had significant variations (*p* < 0.05) between the soil samples. The control soil registered the highest P content (21.48%). In *C. aemula* plants growing at T_1_ and T_2_, low values of total P were found. In contrast, high values of phosphorus were determined for *A. glomeratus* at both transects.

### 3.2. Rhizosphere Minerals

Minerals identified in samples from the pioneer plants’ rhizospheres are listed in Appendix A (Appendix A). In the rhizosphere of the *C. aemula* plant, 96 different minerals were identified. Cristobalite was the main mineral detected with high intensity peaks (Appendix A). In addition, albite, eglestonite, labradorite and quartz were detected but less frequently. With respect to *A. glomeratus*, 108 different minerals were identified. Albite, cristobalite and labradorite were the most frequently detected minerals and showed high intensity X-ray peaks (Appendix A).

### 3.3. Rhizosphere Metals 

In the rhizosphere samples from the pioneer plants *C. aemula* and *A. glomeratus* growing inside the El Chichón volcano, an important gamma of metals was detected (Appendix A). Alkali metals (Na, K), alkaline earth metals (Ba, Ca, Mg, Sr), transition metals (Al, Cd, Co, Cr, Cu, Fe, Mn, Ni, Pb, V, Zn) and a metalloid (As) were found. In both plants, the most abundant metals (above 500 mg kg^−1^) were Al, Ca, Fe, K, Mg and Na. In the *C. aemula* rhizosphere, the metal with the highest concentration value was Fe (8965.33 mg kg^−1^). In the case of the *A. glomeratus* plant, Ca was detected in a higher quantity (4631.67 mg kg^−1^). In addition, Cd was detected in a lower concentration in both plant species.

### 3.4. Bacterial Community Characteristics Associated to Pioneer Plants

#### 3.4.1. Bacterial Community Structure Associated to Pioneer Plants

According to the analysis based on the 16S rRNA gene sequences, 33 bacterial phyla were identified from *A. glomeratus* and *C. aemula* plant rhizospheres collected inside the volcano (Figure 2). Proteobacteria was the most abundant phylum (>60%) in both samples. Additionally, in both plants, the phylum acidobacteria was the second most abundant (~30%). WPS−2 and chloroflexi were identified with an abundance of >20% for both samples. On the other hand, 23 phyla from the rhizospheres of *A. glomeratus* and *C. aemula* collected outside the volcano were identified (Figure 2). The results showed that the most abundant phyla (>60%) corresponded to proteobacteria, while actinobacteria and acidobacteria were ~20%. The rest of the phyla were identified with <20% relative abundance. The identification of bacterial genera showed 50 different groups from the rhizosphere samples that belonged inside the volcano (Figure 3). In both rhizosphere samples, the genus *Burkholderia* was the most abundant. For *A. glomeratus* rhizosphere, *Burkholderia* showed a relative abundance > 40%, while in *C. aemula* it was > 50%. The genus *Salinispora* showed an abundance > 40% in *C. aemula*, while the rest of the bacterial genera for both samples showed relative abundance < 20%. Outside the volcanic complex, 51 different bacterial genera were identified (Figure 3). Also at this site, the results showed that the most abundant bacterial genus in both samples was *Burkholderia*, with relative abundance > 60%, while the rest of the bacterial genera showed an abundance <20%.

#### 3.4.2. Principal Component Analysis

A PCA was used to understand further the contribution of the variables pH, EC, CEC, SOM, C, P, N and C/N ratio and their relationships with the relative abundances of the different bacterial phyla (Figure 4A,B). In the PCA of inside (Figure 4A), the abundance of the bacterial phyla from *C. aemula* was positively correlated with the variables pH, CEC, SOM, C, P, N and C/N, but negatively correlated with the variable EC. In this analysis, proteobacteria was the most abundant phylum in rhizospheric soil samples, followed by acidobacteria, while in the opposite quadrant, the bacterial phyla abundance from *A. glomeratus* was positively correlated with the EC variable and negatively correlated with the variables pH, CEC, SOM, C, P, N and C/N. The most abundant phyla from *A. glomeratus* were proteobacteria, followed by acidobacteria and chloroflexi. Likewise, in the PCA of outside (Figure 4B), the bacterial phyla abundance from *C. aemula* was positively correlated with the variables pH, CEC, SOM, C, N and C/N; however, it was negatively correlated with the variables P and EC. In this case, proteobacteria was the most abundant phylum followed by acidobacteria, actinobacteria and planctomycetes, whereas in the opposite quadrant, the bacterial phyla abundance of *A. glomeratus* was positively correlated with the variables P and EC, but negatively correlated with the variables pH, CEC, SOM, C, N and C/N. The most abundant phyla from *A. glomeratus* were proteobacteria, actinobacteria, acidobacteria and chloroflexi.

### 3.5. Diversity and Genetic Identification of Bacterial Isolates 

A total of 174 bacterial strains were isolated from the rhizospheres and plant tissues (endophytes) of pioneer plants *C. aemula* and *A. glomeratus* that grow outside El Chichón volcano (Table 2). Outside the volcano (transect T_1_ and T_2_), 88 strains in *C. aemula* and 86 in *A. glomeratus* were obtained. From the isolated strains in *C. aemula* and *A. glomeratus* 32 ARDRA groups with different genomic profiles were identified. In this way, it was possible to determine the percentage of relative abundance (RA). The highest RA (40.6%) was determined in the rhizospheres of *C. aemula* and *A. glomeratus*. In contrast, the lowest percentage of RA (9.4%) was recorded in plant tissues (endophytes) of both pioneer plants species. Regarding the diversity and abundance of the species in the bacterial community associated with the pioneer plants, the Shannon-Weaver index allowed determining a high richness (d) and diversity (H) in the rhizosphere of the *A. glomeratus* plants (Table 2). The phylogenetic analysis of the 16S rRNA gene sequences of each representative strain selected by ARDRA profiles of *C. aemula* and *A. glomeratus* revealed that the strains belonged to 13 bacterial genera (Table 3). The strains were taxonomically classified within the phyla Actinobacteria (3.0%), Firmicutes (25.0%) and Proteobacteria (72.0%). The isolated strains of *C. aemula* belonged to the genera *Acinetobacter, Arthrobacter, Bacillus, Burkholderia, Cupriavidus, Pantoea, Lysinibacillus, Klebsiella, Pseudomonas, Serratia* and *Stenotrophomonas*. The majority of these bacteria were isolated mainly from the rhizosphere. The isolated strains of *A. glomeratus* belonged to the genus *Bacillus, Acinetobacter, Burkholderia, Enterobacter, Pandoraea, Pseudomonas* and *Serratia*. In this plant species, only the strains CRM-2, CRM-18 and CRM-19 were obtained from plant tissues and the rest of the bacterial isolates were from the rhizosphere. *Bacillus, Burkholderia,* and *Pseudomonas* were the most abundant bacterial genera. *Bacillus* was more abundant in plant tissues (endophytes). *Burkholderia* and *Pseudomonas* were mainly isolated from the rhizosphere of plants.

### 3.6. Potential of Bacterial Strains as Plant Growth Promoters

Strains *Acinetobacter calcoaceticus* CRM-111, *Agrobacterium larrymoorei* EC-34, *Arthrobacter woluwensis* CRM-152, *Bacillus subtilis* CRM-19, *Brevibacillus choshinensis* W12, *Exiguobacterium indicum* AOB127, *Pseudomonas mosselii* CRM-140, and *Sphingobium yanoikuyae* NFB69 were selected to test plant growth promotion activities due to their rapid growth and easy cultivation under laboratory conditions (Table 4). All strains had the capacity to solubilize inorganic phosphate. These strains formed clear zones (solubilization halos) around the colonies and the phosphate solubilization index ranged from 2.25 to 3.78. The strain *Bacillus subtilis* CRM-19 isolated from the plant *A. glomeratus* had the highest value of P solubilization index. In the same way, the strains had the capacity to synthesize indole acetic acid (IAA). CRM-19, CRM-140 and NFB69 stood out for their high production of IAA. In addition, the strains had nitrogenase activity (ARA), which indicated the ability to fix nitrogen. *Agrobacterium larrymoorei* EC-34 (endophyte) isolated from *C. aemula* showed the highest ARA activity. The ACC deaminase activity was present in six isolates, except in CRM-152 and NFB69. Production of siderophores was observed in the strains CRM-11, EC-34, CRM-152, CRM-19, AOB127 and CRM-140. In general, several of the strains had the capacity to produce exopolysaccharides (EPS), except strains CRM-152 and W12 (Table 4).

### 3.7. Plant Growth Promotion Ability of Bacterial Strains on Pepper Plants (Capsicum Chinense)

Biofertilization using selected isolates had a positive effect on the growth and biochemical parameters of *C. chinense* plants (Table 5). *B. subtilis* CRM-19 had the highest positive effect on total plant height, plant weight, root length and total P content (*p* < 0.05) compared to non-inoculated control plants and to those with added chemical fertilizer. Plants inoculated with the *P. mosselii* CRM-140 showed a significant effect (*p* < 0.05) on root weight as well as on chlorophyll and organic carbon content. Stem diameter increased significantly in plants inoculated with strain CRM-140, but the same effect was observed in those plants treated with strain *B. choshinensis* W12 or with *S. yanoikuyae* NFB69. The number of fruits was higher in the plants inoculated with NFB69 compared to the non-inoculated plants. The plants inoculated with the *A. larrymoorei* EC-34 strain showed a significant effect (*p* < 0.05) on the total nitrogen content.

## 4. Discussion

After a volcanic eruption, an ecological process of ecosystem restoration begins, where microorganisms in association with plants play an important role [42]. Despite the extreme environmental conditions at El Chichón volcano, *C. aemula* (Pteridaceae) and *A. glomeratus* (Poaceae) were identified as the first plants that colonized the recent volcanic deposits inside and outside the volcano [19,43]. In soil samples from inside the volcano, the pH values were moderately acidic in ranges between 5.1 to 6.7. Outside the crater (transects T_1_ and T_2_), the pH values were also acidic. From the above, it stands out that the most acidic pH was found in the rhizospheric soil of *A. glomeratus*. Electrical conductivity (EC) showed significant differences between the sampling sites. The ECs in the outer rhizospheric soils were lower than those from inside the volcano. These variations in pH and EC can be attributed to an increase in volcanic activity and the presence of sulfur species ions, excess Cu, Fe, Al, Mn and other heavy metals, which are commonly found in these volcanic deposits [19,22]. In relation to cation exchange capacity (CEC), significant variations were observed between the sampling sites. The soils from the *C. aemula* and *A. glomeratus* plants outside the volcano, in transect T_2_, had high CEC values compared to the other analyzed sampling sites. The above indicates that processes of retention and mobility of nutrients are taking place despite the extreme conditions of high acidity and temperatures of El Chichón crater-lake [18]. Moreover, it was observed that the content of organic C, total N and the C/N ratio in the rhizospheric soil samples increased as the plants moved away from the crater. These parameters are related to fertility and influence soil functionality. Likewise, the P content in the rhizosphere samples are high indicating that a mineralization process is taking place due to microbial activity; also, high P values are characteristic of volcanic soils [44]. The El Chichón volcanic eruption was one of the most important eruptions in the 20th century, wherein pyroclastic flows covered the volcano slopes and destroyed all the forest cover. After eruption, the immediate mineralogical composition of the El Chichón soil was mainly silicates, plagioclases (albite-anortite) and, secondly, ferromagnesians of hornblende and augite types [45]. The results obtained in this study allowed us to have an idea of the current mineral composition in the rhizospheres of *C. aemula* and *A. glomeratus*. In addition, due to the presence or absence of certain minerals, it is possible to obtain valuable information on the geological origin of the soil material, as well as to know the sedimentation environment on which it matured [46,47]. The XRD patterns analyzed in this study for rhizosphere samples showed a dominance of siliceous minerals, with cristobalite as the dominant species. Cristobalite has been described for many silicic and intermediate volcanic rocks, and it is a common component of volcanic mineralogy, e.g., Bezymianny, Rusia [48], Santiaguito, [49], Unzen, Japan [50], Mt. St. Helens (MSH), USA [51], Merapi, Indonesia [52] and Chaitén, Chile [53]. In the *C. aemula* rhizosphere, alabandite, arsenolite, berlinite, graphite and tazheranite were found (with minor intensity peaks); however, these minerals showed faint intensity peaks in *A. glomeratus* rhizosphere samples. Likewise, calcite and falsterite showed higher intensity in *A. glomeratus* rather than in *C. aemula*; the above-mentioned findings indicate that processes of biomineralization are taking place, which may be attributable to microorganisms within the rhizosphere [54]. From the results, we can also deduce that minerals with more intensity surrounding the roots of *C. aemula* comprise the elements Mn, S, As, Al, P, C, Mg, Ca, Na, Si, Cl, Hg, V and K. This is important since we can relate pioneer plants and bacterial functional groups to helping the acquisition of essential nutrients such as P, S, K and Ca [55]. Plants in recently formed and changing soils provide places for the growth and the proliferation of a diverse microbial community [56] where bacteria interactions establish and maintain core microbiota in the rhizobiome; even in the presence of different minerals, the host plant assembles specific bacterial communities [57]. Our findings and other studies [58,59] allow us to suggest that microbial communities can develop thanks to minerals present in soils, according to their mineralogy, nutritive content and weatherability. In biological weathering by roots, microorganisms play an indispensable role in maintaining a continuous supply of inorganic nutrients for plants [60]. 

At present, there are no studies focused on understanding if there is a preference by *A. glomeratus* and *C. aemula* for certain types of rhizosphere minerals. We believe that the variability of minerals in each plant depends on its physiological and morphological characteristics. *A. glomeratus* species have fibrous type root systems that form aggregates with the soil whereas the *C. aemula* plant is a lithotropic fern that grows mainly on rocky soil that is derived from pyroclastic or tephra material near the muddy littoral zone of the lake. This type of fern has a thin rhizome, which penetrates the stony material and also forms aggregates with the muddy material where this species colonizes. The fact that *A. glomeratus* roots grow below the soil surface may be the reason why it is in contact with a greater number of minerals, while *C. aemula* that grows on volcanic rocks can only be in contact with surface minerals, which are subject to atmospheric events [61]. The rhizospheric samples from El Chichón volcano crater-lake are mainly composed of macronutrients (C, N, P, K), secondary elements (Ca, Mg and Na), microelements (Al, Ba, Cr, Fe, Mn, Zn and Cu) and heavy metals (Pb, Cd, Co, Ni and As). The presence of these elements has also been reported for earlier volcanic environments [24,62]. Metal ions are essential for life because some proteins require metals in order to function [63]. Many plant species belonging to the Pteridaceae and Poaceae families have been described for their capability to live in heavy metal polluted soils [64,65,66,67,68,69,70,71,72,73]. Researchers mentioned some root-induced changes at the root–soil interface of a Pteridaceae plant; they observed no changes in soil pH, decreases in soil redox potential and increases in dissolved organic carbon in soils with As (2270 g kg^−1^) [73]. Plants possess different biochemical mechanisms for metal bioavailability. Some microorganisms and Poaceae plant species are able to synthesize siderophores so they can acquire iron in a bioavailable form and metals such as Cd, Cu, Ni, Pb and Zn [74,75]. 

In addition, plant growth hormones and ACC deaminase produced by plant-associated microbes improve plant growth in metal contaminated soils [76,77,78,79]. As can be seen, there is a close relation between plant and microorganisms in nature, and plant-microbe interaction is either beneficial or harmful to plants [80]. Due to the above, the study of plant microbiome or plant microbiota by culture-based and culture-independent approaches is gaining attention nowadays [81,82,83,84,85]. In this study, the bacterial communities belonging to the rhizosphere samples of the *A. glomeratus* and *C. aemula* plants, from inside and outside the volcano, were dominated by proteobacteria. Proteobacteria, actinobacteria, bacteroidetes, acidobacteria and firmicutes have been reported as the main guests of root microbial communities [86,87,88,89]. In perturbed volcanic soils, it has been revealed that proteobacteria often constitutes the dominant microflora followed by acidobacteria [90,91,92,93]. In early ecosystems where nutrients are limited, proteobacteria have developed different strategies to thrive under different stress conditions; bacteria within this phylum have different biochemical strategies such as phototrophy, photoheterotrophy and chemolithotrophy [94]. Acidobacteria is another ubiquitous and abundant phylum in soil [95]. Some authors have mentioned that in early bacterial communities the proportion of this phylum is lower than proteobacteria [96,97], but in a study carried out in Miyake-jima volcanic deposits, the proportion of Acidobacteria increased from 3.5% to 11.7% with vegetation development [1]. According to our results, despite the less favorable fertility conditions inside the volcano, Acidobacteria showed a higher proportion in the rhizosphere samples from *C. aemula* and *A. glomeratus*. In this extreme volcanic environment, plants are influencing the early activity of the rhizospheric bacterial community, allowing them to be the main colonizers of newly exposed volcanic minerals (lava, ash and tephra). The rhizospheric samples also showed a considerable proportion of chloroflexi; this phylotype has been associated to different natural environments such as hot springs [98], hypersaline mats [99], agricultural soils [100], tundra soils [101], lake sediments [102], hydrothermally active sediments [103] and others. The diversity of ecosystems inhabited by chloroflexi suggests its ecological importance in such habitats as it included mesophilic and thermophilic aerobic and anaerobic chemoorganoheterotrophs and photolithoautotrophic bacteria [104]. The PCA analysis relating relative abundance of the different bacterial phylotypes and physicochemical parameters corroborated that the localization of the rhizosphere samples (inside and outside the volcanic complex) exerts a change with respect to the relationship between the parameters and bacterial diversity. The results of this study allow us to understand how the bacterial communities of the pioneer plants *C. aemula* and *A. glomeratus*, growing inside and outside El Chichón, are associated with soil chemical parameters. It is interesting that the bacterial phyla of both plants are dominated by proteobacteria and acidobacteria; however, their microbiota are not governed by the same rhizosphere soil chemical parameters. There is a predilection by *A. glomeratus* for soils with a higher EC (related to the salts/minerals content), compared to *C. aemula* microbiota, which are related to other chemical parameters. Kim et al. [105] reported that the chemical parameters pH and EC are the ones with the greatest correlation and influence on bacterial community structure in various soil types. The EC and pH parameters in soils depend on natural (edaphic, climatic and biological) and anthropogenic factors (related to agriculture, land use changes and others); however, in native soils or in those without intervention by human beings, the chemical parameters depend on the soil texture. Clay soils tend to have a higher amount of cations, while poor-clay soils (sandy soils) have fewer cations; their dynamics will be ruled mainly by pH and soil water (moisture and water retention capacity). Actually, there is not enough information related to pioneer plants in active volcanic environments and there is much less investigation focused on establishing the biological mechanisms that explain how *A. glomeratus* is more related to the presence of salts/minerals in soil. We believe that this correlation is mainly ruled by plant nutritional requirements, considering the current mineral composition of *C. aemula* and *A. glomeratus* rhizospheres. 

In respect of the taxonomic identity of the bacterial isolates, the phylogenetic analysis through 16S rDNA gene sequences showed that the bacterial community isolated from the rhizospheres of *A. glomeratus* and *C. aemula* included the most usual phylogenetic groups in soil: actinobacteria, bacteroidetes, firmicutes and proteobacteria (α, β and γ). These bacterial groups have specialized metabolisms allowing them to associate with pioneer plants, helping their growth and adaptation to harsh conditions [106]. The analysis of the 16S rRNA gene sequences from strains in the rhizospheres and plant tissue of *A. glomeratus* and *C. aemula* corresponding to the outer part of the volcanic complex showed that the dominant isolates belonged to the phyla actinobacteria, firmicutes and proteobacteria. *Acinetobacter, Arthrobacter, Bacillus, Burkholderia, Cupriavidus, Enterobacter, Klebsiella, Lysinibacillus, Pandoraea, Pantoea, Pseudomonas, Serratia* and *Stenotrophomonas* were the genera found. In both rhizospheres outside the volcano, the abundance of gammaproteobacteria was significant, and among them, *Acinetobacter, Pseudomonas* and *Serratia* were isolated in both samples. *Acinetobacter* spp. are reported as biochemically versatile microorganisms of environmental importance: they are able to degrade oil, hydrocarbons and halogenated organic pollutants; synthesize a wide variety of exopolysaccharides and enzymes; and transform heavy metals [107]. Otherwise, Pseudomonas are known as metabolically diverse; they synthesize important metabolites and enzymes and have PGPB properties such as siderophore synthesis, allowing their survival in extreme environments [108,109]. Nitrogen-fixing bacteria in volcanic environments are important due to the fact that they provide fixed nitrogen to the biosphere. 

At El Chichón volcano, *Klebsiella* species were isolated from the *C. aemula* rhizosphere. This genus is widely distributed in roots and soils and has been confirmed as nitrogen-fixing in plants; it also exhibits deaminase activity, indole-3-acetic acid production, phosphate solubilization and heavy metal removal efficiency [110,111,112]. In addition, the relative abundance of bacterial isolates in plant tissues was lowest compared to that in rhizospheres. It is well known that endophytic bacterial populations occur at lower densities and that proteobacteria are the dominant genus [80,113,114]. The interest of isolating endophytic bacteria lies in their biological importance since they can enhance plant health and growth through the exchange of molecules due to interactions between root chemical substances and microbes [115,116]. In addition, endophytic bacteria are related to the diminution of the stress effects of drought, high temperature, nutritional deficiency and phytopathogens on plants [117]. Some authors have reported that harsh environments are of biotechnological importance because they are a source of microbial diversity with PGPB qualities [118,119]. In this study, the strain *B. subtilis* CRM-19 isolated from the tissues of *A. glomeratus* stood out for its ability to solubilize P and synthesize IAA, siderophores and abundant polysaccharides, and it showed ACC deaminase activity. Researchers have reported the PGPB potential of *Bacillus* species isolated from extreme environments, demonstrating that this genus definitely contributes to the improvement of plant nutrition [11,39,120]. The multifunctional qualities promoting plant growth appreciated in the isolated strains from El Chichón volcano point out that bacterial communities in the rhizosphere samples are effective in combating stress conditions in plants, thus mitigating the negative effects produced by heavy metals and other soil contaminants. EPSs allow tolerance to abiotic stress and contributes to the colonization of the root surface; meanwhile, siderophores contribute to the protection of the bacteria against rhizospheric pathogens and they play a key role in mineralization [121]. Nitrogen is not present in soils despite it being the major constituent of the atmosphere compared to the other gases [122]; the conversion of N_2_ to a biologically available form (NH_3_) takes place thanks to the nitrogen-fixing bacterial diversity at El Chichón volcano. This study also shows the plant growth-promoting potential of the isolated strains by an inoculation assay in *C. chinense* plants. All the isolates were able to induce plant growth promotion. The values obtained from the morphometric variables studied showed that the inoculated isolates had a similar growth effect on plants as that obtained in those amended with KNO_3_-N. The strains *B. subtilis* CRM-19, *A. woluwensis* CRM-152 and *P. mosselii* CRM-140 had the highest effect on the total content of phosphorus, nitrogen and organic carbon, respectively. This is very important since these chemical elements are essential for different biochemical processes such as photosynthesis, protein and phospholipid biosynthesis, nucleic acid synthesis, membrane transport, energy transformation and cell division, among others. Higher nutrient absorption leads to a better development of the root system as well as the aerial part of the plants (branching), and therefore, to an accumulation of dry matter [123]. Inoculation of pepper plants with extremophilic PGPB has been reported previously [39,119], where an improvement in shoot growth and a corresponding increase in root biomass in seedlings were observed. Bacteria isolated from the El Chichón volcano improve the nutrient cycle and thus contribute to the fertility of the volcanic soil. Extremophilic bacterial communities in deteriorated environmental conditions improve plant growth and thus reduce environmental contamination, including that produced by poor agricultural practices. It is worth mentioning that future studies are required to prove the nature of the strains for potential use as bioinoculants in agriculture and other applications.

## 5. Conclusions

This study contributes to the knowledge of the rhizospheric and endophytic bacterial communities isolated from *A. glomeratus* and *C. aemula* that grow at El Chichón active volcano. The isolated strains showed important PGPB traits and a positive effect on the growth and development of *C. chinense*. The wide diversity of bacteria associated to pioneer plants with PGPB qualities represent an alternative for the recovery of eroded environments. It is necessary to continue investigating the relationship between physicochemical, biological and genomic factors related to the pioneer plants to understand more about the interactions between plants, soil and microorganisms. This work can be the basis for the study of soil restoration through pioneer plants with biological capacity and stability in extreme environments. Research studies in volcanic areas in Mexico may be an alternative for attending to national needs of soil restoration by using early flora from difficult survival environments.

## Figures and Tables

**Figure 1 microorganisms-10-01568-f001:**
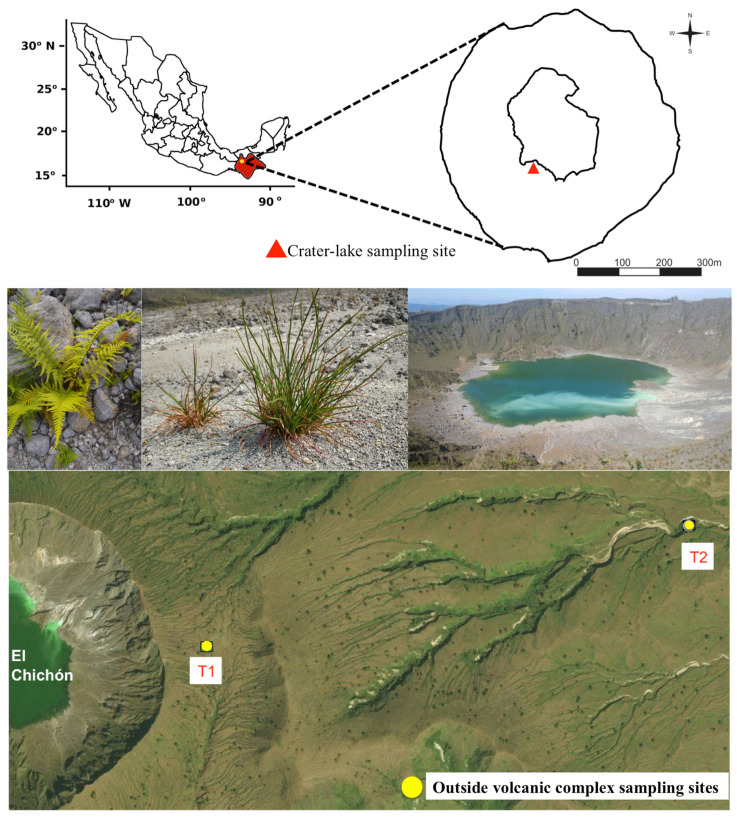
Sampling sites at El Chichón volcano, Chiapas (Mexico).

**Figure 2 microorganisms-10-01568-f002:**
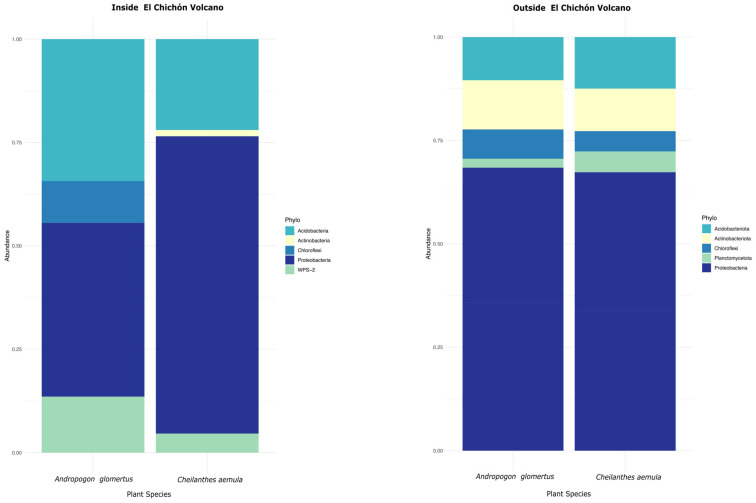
Stacked bar graphics of the relative abundance of the different bacterial phyla found in *A. glomeratus* and *C. aemula* rhizosphere samples, inside and outside the volcanic complex.

**Figure 3 microorganisms-10-01568-f003:**
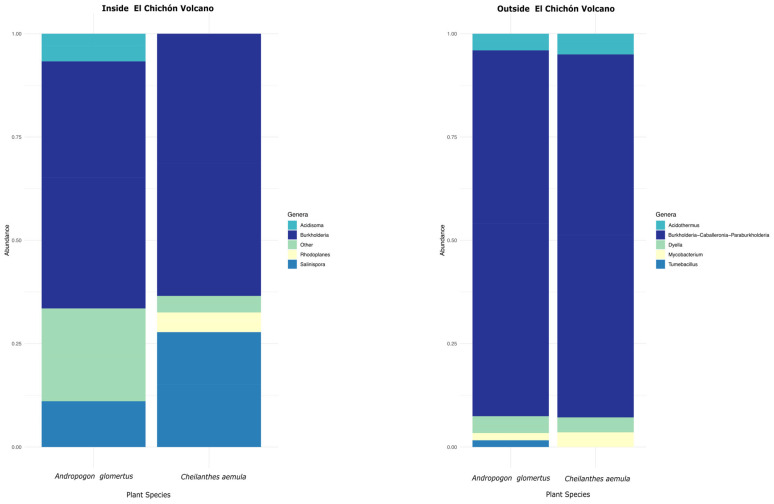
Stacked bar graphics of the relative abundance of the different bacterial genera found in *A. glomeratus* and *C. aemula* rhizosphere samples, inside and outside the volcanic complex.

**Figure 4 microorganisms-10-01568-f004:**
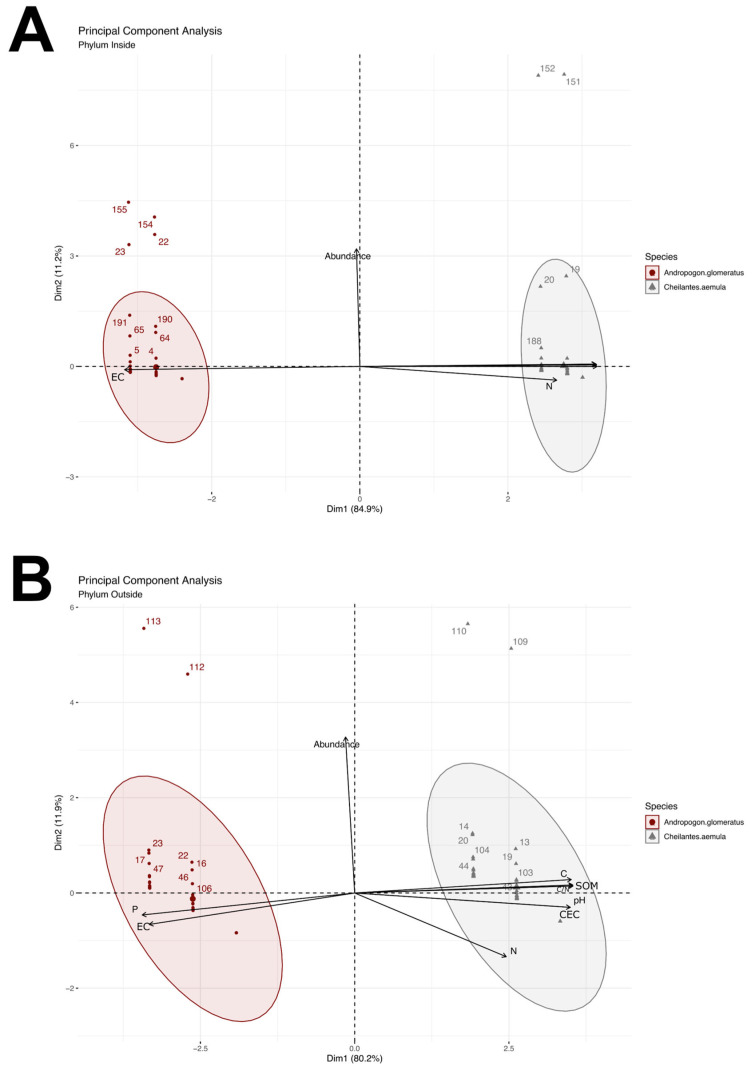
Principal component analysis considering the relative abundance of the bacterial phyla and soil physicochemical characteristics. (**A**) Principal component analysis from samples inside the volcanic complex (crater-lake). (**B**) Principal component analysis from samples outside the volcanic complex.

**Table 1 microorganisms-10-01568-t001:** Physicochemical characteristics of soil samples at El Chichón volcano.

Soil Sample	Sampling Location	pH	EC ^†^ (dSm^−1^)	CEC ^‡^ (Cmol kg^−1^)	Total N (%)	Organic C (%)	Total P (%)	C:N Ratio
*C. aemula* Rhizosphere (Inner crater-lake) [19]	17° 36′ 08″ N 93° 23′ 14″ W	6.7 A ^¶^	0.48 C	16.7 C	0.19 D	1.74 F	2.25 F	9.10 C
*A. glomeratus* Rhizosphere (Inner crater-lake) [19]	17° 36′ 08″ N 93° 23′ 14″ W	5.1 C	0.97 A	12.0 CD	0.17 D	1.28 F	1.06 G	7.50 D
*C. aemula* Rhizosphere (Outer crater-lake) [This study]	17° 36′ 08″ N 93° 22′ 32″ W (Transect 1)	5.4 BC	0.02 D	7.8 D	0.23 C	3.22 C	3.66 E	14.0 B
*A. glomeratus* Rhizosphere (Outer crater-lake) [This study]	17° 36′ 08″ N 93° 22′ 32″ W (Transect 1)	4.8 CD	0.02 D	4.1 E	0.21 C	3.06 D	17.10 B	15.0 A
*C. aemula* Rhizosphere (Outer crater-lake) [This study]	17° 36′ 55″ N 93° 20′ 96″ W (Transect 2)	6.2 AB	0.46 C	27.0 A	0.60 A	9.40 A	4.50 D	15.60 A
*A. glomeratus* Rhizosphere (Outer crater-lake) [This study]	17° 36′ 55″ N 93° 20′ 96″ W (Transect 2)	5.8 B	0.58 C	23.0 B	0.50 B	7.20 B	11.30 C	14.40 B
Control Soil [This study]	17° 36′ 08″ N 93° 23′ 14″ W	4.3 D	0.91 B	5.17 DE	0.16 D	2.94 E	21.48 A	10.30 C
*p*-value		0.00	0.00	0.00	0.00	0.00	0.00	0.00
HSD^#^ (*p* < 0.05)		0.298	0.186	2.019	0.090	0.938	1.392	5.053

^†^ EC: Electrical conductivity; ^‡^ CEC: Cation exchange capacity; ^¶^ mean values of three replicates. Means followed by the same letter are non-significant (Tukey test, *p* < 0.05); ^#^ HSD: Honest Significant Difference.

**Table 2 microorganisms-10-01568-t002:** Diversity and abundance of rhizosphere and endophytic bacterial species isolated from pioneer plants outside El Chichón volcano.

Pioneer Plant (Isolate Section)	No. of Isolates	No. of Groups ARDRA Profiles ^a^	Relative Abundance (%)	Shannon-Weaver Index ^b^	
				Richness (*d*)	Diversity (*H*)
*Cheilantes aemula* (Rhizosphere)	55	13	40.6	3.13	2.84
*Cheilantes aemula* (Endophytes)	33	3	9.4	2.14	2.25
*Andropogon glomeratus* (Rhizosphere)	51	13	40.6	3.22	2.43
*Andropogon glomeratus* (Endophytes)	35	3	9.4	2.26	1.38
Total	174	32	100		

^a^ ARDRA profiles, amplified rDNA restriction analysis obtained with AluI restriction enzyme (AG^CT); ^b^ Shannon–Weaver index. It was estimated using the method reported by López-Fuentes et al. [31].

**Table 3 microorganisms-10-01568-t003:** Phylogenetic affiliation of bacterial strains isolated from the pioneer plants growing at the El Chichón volcano.

Representative Isolate	Closest-NCBI Match/Similarity (%) ^a^	Accession Number	Pioneer Plant	Plant Isolate Section	Sampling Site ^b^	Phylum
CRM-9	*Bacillus pumilus* CP1/99	OL843132	*C. aemula*	Endophytes	OC-(T_1_)	Firmicutes
CRM-20	*Pantoea ananatis* RB163/99	OL843136	*C. aemula*	Endophytes	OC-(T_1_)	Proteobacteria
CRM-125	*Arthrobacter* sp. HPG166/99	OL843147	*C. aemula*	Rhizosphere	OC-(T_1_)	Actinobacteria
CRM-14	*Bacillus altitudinis* SR6-1/97	OL843133	*C. aemula*	Endophytes	OC-(T_2_)	Firmicutes
CRM-95	*Acinetobacter seifertii* 34M/100	OL843139	*C. aemula*	Rhizosphere	OC-(T_2_)	Proteobacteria
CRM-153	*Acinetobacter* sp. DB4/100	OL843157	*C. aemula*	Rhizosphere	OC-(T_2_)	Proteobacteria
CRM-152	*Arthrobacter woluwensis* SCC8/99	OL843156	*C. aemula*	Rhizosphere	OC-(T_2_)	Firmicutes
CRM-98	*Burkholderia cepacia* JCM 5511/99.4	OL843140	*C. aemula*	Rhizosphere	OC-(T_2_)	Proteobacteria
CRM-183	*Burkholderia cepacia* JCM 5511/99	OL843161	*C. aemula*	Rhizosphere	OC-(T_2_)	Proteobacteria
CRM-93	*Cupriavidus* sp. WS/99.9	OL843138	*C. aemula*	Rhizosphere	OC-(T_2_)	Proteobacteria
CRM-147	*Klebsiella michiganensis* DMQ7 /100	OL843154	*C. aemula*	Rhizosphere	OC-(T_2_)	Proteobacteria
CRM-148	*Lysinibacillus fusiformis* PTS4/99	OL843155	*C. aemula*	Rhizosphere	OC-(T_2_)	Firmicutes
CRM-194	*Pseudomonas putida* JYR-1/100	OL843162	*C. aemula*	Rhizosphere	OC-(T_2_)	Proteobacteria
CRM-120	*Pseudomonas* sp. RABA8/100	OL843146	*C. aemula*	Rhizosphere	OC-(T_2_)	Proteobacteria
CRM-90	*Serratia* sp. BR13897/99	OL843137	*C. aemula*	Rhizosphere	OC-(T_2_)	Proteobacteria
CRM-117	*Stenotrophomonas maltophilia* AB5-SW2/100	OL843145	*C. aemula*	Rhizosphere	OC-(T_2_)	Proteobacteria
CRM-2	*Bacillus pumilus* AM08/93	OL843131	*A. glomeratus*	Endophytes	OC-(T_1_)	Firmicutes
CRM-18	*Bacillus safensis* IBK-4/100	OL843134	*A. glomeratus*	Endophytes	OC-(T_1_)	Firmicutes
CRM-19	*Bacillus subtilis* SR3-4 /100	OL843135	*A. glomeratus*	Endophytes	OC-(T_2_)	Firmicutes
CRM-111	*Acinetobacter calcoaceticus* 41/100	OL843142	*A. glomeratus*	Rhizosphere	OC-(T_1_)	Proteobacteria
CRM-128	*Bacillus* sp. NCIM 5035/99	OL843148	*A. glomeratus*	Rhizosphere	OC-(T_1_)	Firmicutes
CRM-129	*Burkholderia contaminans* PK5-6/100	OL843149	*A. glomeratus*	Rhizosphere	OC-(T_1_)	Proteobacteria
CRM-114	*Burkholderia* sp. C3B1M/100	OL843144	*A. glomeratus*	Rhizosphere	OC-(T1)	Proteobacteria
CRM-167	*Burkholderia* sp. LRSZN43/100	OL843160	*A. glomeratus*	Rhizosphere	OC-(T_2_)	Proteobacteria
CRM-163	*Enterobacter tabaci* cjy13/100	OL843158	*A. glomeratus*	Rhizosphere	OC-(T_2_)	Proteobacteria
CRM-113	*Pandoraea sputorum* NCTC13161/100	OL843150	*A. glomeratus*	Rhizosphere	OC-(T_2_)	Proteobacteria
CRM-130	*Pandoraea* sp. LMG 31010/100	OL843143	*A. glomeratus*	Rhizosphere	OC-(T_2_)	Proteobacteria
CRM-140	*Pseudomonas mosselii* NG1/100	OL843153	*A. glomeratus*	Rhizosphere	OC-(T_2_)	Proteobacteria
CRM-165	*Pseudomonas putida* L28676.1/100	OL843159	*A. glomeratus*	Rhizosphere	OC-(T_2_)	Proteobacteria
CRM-135	*Pseudomonas* sp. TNT7/100	OL843151	*A. glomeratus*	Rhizosphere	OC-(T_2_)	Proteobacteria
CRM-136	*Serratia marcescens* WVU-010/98	OL843152	*A. glomeratus*	Rhizosphere	OC-(T_2_)	Proteobacteria
CRM-110	*Serratia marcescens* WVU-010/100	OL843141	*A. glomeratus*	Rhizosphere	OC-(T_2_)	Proteobacteria

^a^ Similarity percentage was estimated by considering the number of nucleotide-substitutions between a pair of sequences divided by the total number of compared bases × 100%; ^b^ sampling site: OC-T_1_ = Outer crater (transect T_1_); OC-T_2_ = Outer crater (transect T_2_).

**Table 4 microorganisms-10-01568-t004:** Plant growth promotion activities in bacterial strains isolated from the pioneer plants growing at El Chichón volcano.

Treatments	P Solubilization Index	IAA (mg L^−1^)	ARA ^¥^	ACC Deaminase	Siderophore	EPS
*Acinetobacter calcoaceticus* CRM-111	2.58 ± (0.42)	9.90 ± (0.78)	112.8 ± (9.1)	+	+	+
*Agrobacterium larrymoorei* EC-34	2.79 ± (0.20) ^≠^	11.47 ± (0.74)	311.1 ± (7.4)	+	+	+
*Arthrobacter woluwensis* CRM-152	2.25 ± (0.42)	10.40 ± (0.92)	207.6 ± (8.0)	−	+	−
*Bacillus subtilis* CRM-19	3.78 ± (0.25)	17.10 ± (0.61)	106.6 ± (5.2)	+	+	+
*Brevibacillus choshinensis* W12	3.13 ± (0.44)	9.47 ± (1.05)	117.9 ± (2.7)	+	−	−
*Exiguobacterium indicum* AOB127	3.12 ± (0.92)	10.47 ± (1.07)	149.5 ± (6.1)	+	+	+
*Pseudomonas mosselii* CRM-140	2.83 ± (0.80)	15.60 ± (1.11)	162.7 ± (5.5)	+	+	+
*Sphingobium yanoikuyae* NFB69	3.53 ± (0.53)	12.83 ± (1.89)	189.1 ± (7.7)	−	−	+

+: positive activity; −: negative activity; ^≠^ mean values of three replicates. The values in parentheses are standard deviations; ^¥^ ARA, acetylene reduction assay (nmol C_2_H_4_ per culture fresh weigh h^−1^).

**Table 5 microorganisms-10-01568-t005:** Growth parameters for *Capsicum chinense* plants inoculated with PGP bacteria isolated from pioneer plants growing in El Chichón volcano.

Treatment	Total Height (cm)	Plant Weight (g)	Root Weight (g)	Root Length (cm)	Stem Diameter (mm)	Number Fruits	Clorophyll (mg mL^−1^)	Total P (%)	Total N (%)	Organic C (%)
*A. calcoaceticus* CRM-111	76.0 B	4.63 B	1.7 E	39.5 CD	5.3 AB	2.83 AB	3.18 B	0.27 D	1.42 C	51.6 F
*A. larrymoorei* EC-34	71.5 CD ^¥^	3.9 DE	1.9 CDE	36.5 DE	4.3 CD	2.3 BC	2.24 E	0.30 C	1.47 A	51.9 E
*A. woluwensis* CRM-152	76.3 B	4.6 B	1.3 F	35.6 DE	5.0 BC	2.0 BC	1.88 F	0.29 C	1.48 A	49.8 H
*B. subtilis* CRM-19	84.8 A	5.3 A	2.1 BC	49.6 A	5.0 BC	2.6 AB	2.72 C	0.37 A	1.29 G	52.4 D
*B. choshinensis* W12	74.3 BCD	4.3 BC	1.3 F	43.3 ABC	6.0 A	2.66 AB	2.50 D	0.29 C	1.37 D	49.4 I
*E. indicum* AOB127	75.6 BC	4.2 CD	2.0 BCD	31.8 EF	5.0 BC	1.0 D	2.25 E	0.34 B	1.32 E	52.8 C
*P. mosselii* CRM-140	76.5 B	5.0 A	2.6 A	41.6 BCD	6.0 A	2.5 AB	3.57 A	0.30 C	1.30 F	54.1 A
*S. yanoikuyae* NFB69	82.8 A	4.9 B	2.2 B	47.1 AB	6.0 A	3.33 A	3.34 B	0.30 C	1.46 B	54.0 B
Chemical fertilizer	70.5 D	3.8 E	1.8 DE	32.6 EF	4.0 D	2.33 BC	1.36 G	0.24 E	1.29 G	51.0 G
Uninoculated	65.5 E	3.7 E	1.7 E	26.8 F	4.0 D	1.5 CD	0.93 H	0.23 F	1.22 H	47.3 J
*p*-value	0.000	0.000	0.000	0.000	0.000	0.000	0.000	0.000	0.000	0.000
HSD ^£^ (*p* < 0.05)	4.219	0.325	0.215	6.510	0.961	0.955	0.177	0.0093	0.0060	0.0086

^¥^ Mean values of six replicates. Means followed by same letter are non-significant (Tukey test, *p* < 0.05); ^£^ HSD: Honest Significant Difference.

## Data Availability

The authors declare that all relevant data supporting the findings of this study are included in this article.

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
