# Peer review of "Bacterial Community with Plant Growth-Promoting Potential Associated to Pioneer Plants from an Active Mexican Volcanic Complex"

_microorganisms, 2022, doi:10.3390/microorganisms10081568_

Round 1

Reviewer 1 Report

The manuscript presents very interesting studies concerning the analysis of microbial communities associated with pioneer plants from the active volcanic complex. The manuscript is well structured and written, it contains a really good piece of scientific work. However, I have some questions and it would be advisable to make some corrections before accepting:

  1. Introduction section– a valuable addition would be to provide more information about the importance of studying the bacterial communities from extreme environments and the advantage of the metagenomic approach over the traditional microbiology techniques. 
  2. Line 123 – please provide the full details about the primers. 
  3. Line 131 – which version of Greengenes was used? 
  4. Figure 2 and Figure 3 is unreadable. Please increase the font. I suggest grouping low abundant taxa (e.g. < 5% abundance) as one group of “others”.

Author Response

 (Reviewer 1)

Comments and Suggestions for Authors

The manuscript presents very interesting studies concerning the analysis of microbial communities associated with pioneer plants from the active volcanic complex. The manuscript is well structured and written, it contains a really good piece of scientific work. However, I have some questions and it would be advisable to make some corrections before accepting:

Point 1: Introduction section– a valuable addition would be to provide more information about the importance of studying the bacterial communities from extreme environments and the advantage of the metagenomic approach over the traditional microbiology techniques. 

Response 1: We appreciate the reviewer's observations and comments on our work. We have carefully reviewed the manuscript and made the modifications suggested by the reviewer for a better understanding.

In the Introduction section, we have provided more information related to the extreme environment bacterial community. Also, we have added information on the advantages of metagenomic analysis.

Point 2: Line 123 – please provide the full details about the primers. 

Response 2:   The primers that were used for the amplification of V3–V4 bacterial 16S rRNA variable región were:

Bakt_341F: CCTACGGGNGGCWGCAG and Bakt_805R: GACTACHVGGGTATCTAATCC

Point 3: Line 131 – which version of Greengenes was used? 

Response 3:The version of the Greengenes software used in this study is available from http://greengenes.lbl.gov/

Point 4: Figure 2 and Figure 3 is unreadable. Please increase the font. I suggest grouping low abundant taxa (e.g. <5% abundance) as one group of “others”.

Response 4: We agree with the reviewer. We have increased the quality and resolution of the images in figures 2 and 3.

Reviewer 2 Report

The manuscript “Rhizospheric and Endophytic Bacterial Community with Plant Growth-Promoting Potential Associated to Pioneer Plants from an Active Mexican Volcanic Complex” by Rincón-Molina and colleagues describes the diversity and PGPB potential of the bacterial community inhabiting in the rhizosphere and endosphere of two pioneer plant species grown inside and outside the El Chichón volcano. In addition, the authors characterized physiochemically the rhizosphere soils and the content of minerals and metals of those soils.

Overall, the article is very well written and presents some interesting data. However, I have several concerns that I would like that the authors reflect on:

-       Is the work conducted with endophytic bacteria significant to be reflected in the title?

-       why there is no PCA analysis done with all the physicochemical data and all the plants from the different locations?

-       Why was the plant growth promotion ability of selected strains tested on pepper plants (Capsicum chinense)?

Minor aspects:

Figure 1- Why the local site for plants collection inside volcano is now shown?

Line 205- Why the authors claim that “The rhizospheric soil pH from plants inside the volcano was moderately acidic (5.1 to 6.7) compared to the soil pH from outside the volcanic complex.”

Table 1- Where was the control soil collected from?

Line 244-263 It is difficult to follow the text. For instance, “Proteobacteria was the most abundant phylum (>60%) in both samples. In A. glomeratus the phylum acidobacteria had >60% relative abundance, while in C. aemula this phylum was represented with <50%.” So, if proteobacteria is the most abundant phylum in both samples, how the phylum acidobacteria can have a relative abundance >60%? Please, explain it better and revise the text.

Figure 2 and 3- It is difficult to follow the relative abundance of each phylum or genus with the colour range selected. They need to be reformulated.

Table 4- How EPS were measured? There is no description in the M&M section

Table 5- It is difficult to understand why the pepper treatment with chemical fertilizer is similar to the uninoculated treatment in most of the parameters analyzed. Is there any explanation?

All over the text, genus and species names should in italic.

Author Response

(Reviewer 2)

Comments and Suggestions for Authors

Point 1: The manuscript “Rhizospheric and Endophytic Bacterial Community with Plant Growth-Promoting Potential Associated to Pioneer Plants from an Active Mexican Volcanic Complex” by Rincón-Molina and colleagues describes the diversity and PGPB potential of the bacterial community inhabiting in the rhizosphere and endosphere of two pioneer plant species grown inside and outside the El Chichón volcano. In addition, the authors characterized physiochemically the rhizosphere soils and the content of minerals and metals of those soils.

Overall, the article is very well written and presents some interesting data. However, I have several concerns that I would like that the authors reflect on:

Response 1:  We appreciate the reviewer's comments and observations on our work. We have reviewed each of the questions or doubts and have given the clarifications and correct answers.

Point 2: Is the work conducted with endophytic bacteria significant to be reflected in the title?

Response 2:  We agree with the reviewer. We have changed the title of the work “Rhizospheric and Endophytic Bacterial Community with Plant Growth-Promoting Potential Associated to Pioneer Plants from an Active Mexican Volcanic Complex” by “Bacterial Community with Plant Growth-Promoting Potential Associated to Pioneer Plants from an Active Mexican Volcanic Complex”.

Point 3: why there is no PCA analysis done with all the physicochemical data and all the plants from the different locations?

Response 3:  The PCA analysis was only done on the pioneer plants Cheilantes aemulaand Andropogon glomeratus, since they are the only plants that grow inside the volcanic complex and at the sampling sites (Figure 1). Then, a PCA was carried out to better understand the contribution of the physicochemical variables pH, EC, CEC, SOM, C, P, N and the C/N ratio on the relative abundance at the bacterial phylum level that make up the bacterial communities associated with these two species of plants.

Point 4: Why was the plant growth promotion ability of selected strains tested on pepper plants (Capsicum chinense)?

Response 4: 

The strains Acinetobacter calcoaceticusCRM-111, Agrobacterium larrymooreiEC-34, Arthrobacter woluwensisCRM-152, Bacillus subtilisCRM-19, Brevibacillus choshinensisW12, Exiguobacterium indicumAOB127, Pseudomonas mossseliiCRM-140 and Sphingobium yanoikuyaeNFB69 were selected for use as biofertilizers on pepper plants (Capsicum chinense), because they stand out for their potential as PGP and for their high capacity to fix nitrogen and solubilize phosphate. Furthermore, some of these bacterial species, such as Acinetobacter calcoaceticus, Agrobacterium larrymoorei (Rhizobium),Bacillus subtilis, and Pseudomonas mossseliihave been reported as excellent PGPB.

Minor aspects:

Point 5: Figure 1- Why the local site for plants collection inside volcano is now shown?

Response 5:Rhizospheric soil samples were collected from the interior of the volcano to determine the content of metals, minerals and a metagenomic study.

Point 6: Line 205- Why the authors claim that “The rhizospheric soil pH from plants inside the volcano was moderately acidic (5.1 to 6.7) compared to the soil pH from outside the volcanic complex.”

Response 6 :  We agree with the reviewer. In Line 205, we have changed the paragraph "The rhizospheric soil pH from plants inside the volcano was moderately acidic (5.1 to 6.7) compared to the soil pH from outside the volcanic complex. The registered pH value of the control soil was the most acidic (4.3)by “The rhizospheric soil pH from plants inside the volcano was moderately acidic (5.1 to 6.7) compared to the soil pH from transect T1(outside the volcanic complex) where the pH was strongly acidic (range 5.4 to 4.8). The recorded pH value of the control soil was the most acidic (4.3)”.

Point  7: Table 1- Where was the control soil collected from?

Response 7:  In M & M section, 2.2.1. Physicochemical analysis, we add the following information: “The same determinations were done to a soil sample used as control, which was collected inside the volcanic complex. The control soil consisted of a soil without plants.

Point 8: Line 244-263 It is difficult to follow the text. For instance, “Proteobacteria was the most abundant phylum (>60%) in both samples. In A. glomeratus the phylum acidobacteria had >60% relative abundance, while in C. aemula this phylum was represented with <50%.” So, if proteobacteria is the most abundant phylum in both samples, how the phylum acidobacteria can have a relative abundance >60%? Please, explain it better and revise the text.

Response 8:  We agree with the reviewer. We have revised the text of the paragraph in lines 244-263 and have written and improved the grammatical structure of the paragraph for better understanding.

Point 9: Figure 2 and 3- It is difficult to follow the relative abundance of each phylum or genus with the colour range selected. They need to be reformulated.

Response 9:  We agree with the reviewer. We have revised and reformulated the images (colour range) in figures 2 and 3, for a better observation of the relative abundance of each bacterial phylum or genus.

Point 10: Table 4- How EPS were measured? There is no description in the M&M section

Response 10:  We agree with the reviewer. In the M&M section, we have added the technique for EPS measurement :

2.4.6. Exopolysaccharide (EPS)production

EPS was determined as described by Paulo et al. (40). The evaluated strains were inoculated into sterilized filter paper discs of 5 mm Ø placed on the surface of culture medium PY modified containing (2% yeast extract, 1.5% K2HPO4, 0.02% MgSO4, 0.0015% MnSO4, 0.0015% FeSO4, 0.003% CaCl2, 0.0015% NaCl, 1.5% agar and 10% sucrose, with pH adjusted to 7.0) followed by incubation at 30 °C for 48 h. A mucoid layer formed around the paper discs suggested EPS production. In order to confirm the presence of EPS, the mucoid layer was transferred to a tube containing 2 mL absolute ethanol. The EPS presence was confirmed by the formation of a precipitate.

Point 11: Table 5- It is difficult to understand why the pepper treatment with chemical fertilizer is similar to the uninoculated treatment in most of the parameters analyzed. Is there any explanation?

Response 11:  We appreciate the comment. According to Tukey's test (p<0.05), no significant difference was observed between plants treated with chemical fertilizer and plants without inoculation. This result can be attributed to the biological and physiological nature of the plant and its nutritional needs. Also, although the plants grew in an inert substrate (peat moss plus agrolite), they were able to obtain few nutrients from the mineralization process of decomposing organic matter.

Point 12: All over the text, genus and species names should in italic.

Response 12:  We agree with the reviewer. The names of the genus and species have been correctly written in italics.